# Stability Study of Parenteral N-Acetylcysteine, and Chemical Inhibition of Its Dimerization

**DOI:** 10.3390/ph16010072

**Published:** 2023-01-03

**Authors:** Nicolas Primas, Guillaume Lano, Damien Brun, Christophe Curti, Marion Sallée, Emmanuelle Sampol-Manos, Edouard Lamy, Charleric Bornet, Stéphane Burtey, Patrice Vanelle

**Affiliations:** 1Service Central de la Qualité et de L’information Pharmaceutiques (SCQIP), Pharmacy Department, Assistance Publique—Hôpitaux de Marseille (AP-HM), 13005 Marseille, France; 2CNRS, Institut de Chimie Radicalaire ICR, UMR 7273, Equipe de Pharmaco-Chimie Radicalaire, Aix-Marseille University, 13385 Marseille, France; 3Centre of Nephrology and Renal Transplantation, Hôpital de la Conception, AP-HM, 13005 Marseille, France; 4Pharmacokinetics Department, Assistance Publique-Hôpitaux Marseille (AP-HM), 13005 Marseille, France; 5UMR 7287 CNRS, Institut des Sciences du Mouvement ISM, Faculté des Sciences du Sport, Aix-Marseille University, 13385 Marseille, France; 6Pharmacie Usage Intérieur Hôpital Conception, Assistance Publique—Hôpitaux de Marseille (AP-HM), Hôpital de la Conception, 13005 Marseille, France; 7INSERM, Centre de Recherche en Cardiovasculaire et Nutrition (C2VN), Institut National de la Recherche pour l’Agriculture, l’Alimentation et l’Environnement (INRAE), Aix-Marseille University, 13385 Marseille, France

**Keywords:** N-acetylcysteine, stability study, parenteral administration

## Abstract

Parenteral N-acetylcysteine has a wide variety of clinical applications, but its use can be limited by a poor chemical stability. We managed to control parenteral N-acetylcysteine stability, and to study the influence of additives on the decrease of N-acetylcysteine degradation. First, an HPLC-UV dosing method of N-acetylcysteine and its main degradation product, a dimer, was validated and the stability without additive was studied. Then, the influence of several additives (ascorbic acid, sodium edetate, tocopherol and zinc) and of temperature on N-acetylcysteine dimerization was evaluated. Finally, the influence of zinc gluconate at different concentrations (administrable to patients) was investigated. Zinc gluconate at 62.5 µg·mL^−1^ allows the stabilization of 25 mg·mL^−1^ N-acetylcysteine solution for at least 8 days when stored at 5 ± 3 °C.

## 1. Introduction

N-acetylcysteine (NAC) can be prescribed intravenously as a treatment for acute paracetamol toxicity [1]. It is also under investigation for a wide variety of other indications, such as severe influenza [2], as an adjunct to pharmaco-invasive reperfusion in patients presenting early after a ST-segment elevation myocardial infarction [3] or for the decrease of the oxidative stress during chronic kidney disease [4].

Its stability was previously investigated at ambient temperature, in dextrose 5% in water (D5W), at 26 mg·mL^−1^ in PVC containers [5]. In this study, NAC at ambient temperature was stable for 60 h but not for 72 h (the authors observed more than 10% degradation). When stored at ambient temperature, 60 mg·mL^−1^ NAC compounded injectable solutions were stable for 72 h in another study, both in D5W%, normal saline (NS) and 0.45% sodium chloride [6]. For inhalation or for oral administration only, 200 mg·mL^−1^ NAC compounded solution in sterile water for injection is described in United States Pharmacopeia, with edetate disodium hydrate to avoid oxidation and sodium hydroxide to adjust pH. The beyond-use-date was defined at 60 days after compounding. The compounding procedure emphasizes the necessity to completely fill the container to minimize the amount of oxygen concentration [7].

Beyond-Use-Date of commercial drugs must be established with a stability study designed as required by the scientific guideline established by the International Council for Harmonization (ICH) [8], using a dosing method validated to be stability-indicating [9]. A stability-indicating method was defined as a validated quantitative analytical method that can detect the changes with time in the properties of the drug product without interference [10].

In our hospital, parenteral NAC is currently under evaluation at 25 mg·mL^−1^ in D5W (qs 80 mL) for a new clinical application [4]. We decided first to confirm the published stability, and we tried to improve it with chemical additives administrable to patients.

Our preliminary results showed that NAC major degradation product yielded from NAC oxidation. There is a wide variety of available antioxidizing agents, but to avoid high risk compounding steps with raw material, we decided to select only antioxidizing agents available as ready-to-use commercial drugs: ascorbic acid, zinc gluconate and tocopherol. Doses were tried first to correspond to the adjunction of one commercial drug vial to NAC to test the higher antioxidant property as possible. Disodium edetate was also tested at 2 mg·mL^−1^ as undiluted commercial NAC drug contain this excipient at the same concentration. From these results, we planned to find the best antioxidizing agent and next to decrease dose to determine the lowest dose as possible to be compatible with human administration.

## 2. Results

### 2.1. N-acetylcysteine Dosing Method Validation

A 25 mg·mL^−1^ stock solution of NAC was prepared: 1.25 mL of commercial 200 mg·mL^−1^ NAC were aliquoted and diluted in 10 mL of mobile phase.

Linearity of HPLC-UV standard curve for NAC was determined with seven concentrations ranging from 125 to 1500 µg·mL^−1^ prepared in sextuplicate. Linearity was proved between 250 and 1500 µg·mL^−1^ (R^2^ > 0.99 and deviation < 10% for each concentration). Repeatability, intermediate precision, accuracy, and uncertainty were determined at three levels of concentration 450 µg·mL^−1^, 500 µg·mL^−1^ and 550 µg·mL^−1^ corresponding to the level of theoretical dilution and the conformity margins (±10%). A robustness study was also conducted with slight variations on several parameters (mobile phase composition and pH, flow, wavelength and column). pH increase of mobile phase and column furnisher were shown to have an influence on reliability of the results (%RSD > 2.0%). A chromatogram of a 500 µg·mL^−1^ NAC solution is showed in Figure 1 and results are reported in Table 1 and Table 2.

To determine NAC content in 25 mg·mL^−1^ parenteral solution, 200 µL were diluted qs 10 mL of mobile phase and vortexed 1 min to obtain a theoretical 500 µg·mL^−1^ N-acetylcysteine solution.

### 2.2. Forced Degradation Studies

Forced degradation studies intend to demonstrate the stability indicating character of the chromatographic method. A stock solution of NAC was exposed to several stressed conditions including intensive heating, oxidation, light, and extremes pH values. Results are reported in Table 3.

The last column of Table 3 presents the retention times of degradation products (each retention time correspond to one degradation product). Observed chromatograms can be found in Appendix A.

A 3% decrease in NAC content under light irradiation (sunlamp, 4 weeks) and a 24% decrease under heating conditions (80 °C, 3 h) were noticed. Under acidic (HCl 0.5 M, 1 min) and basic (NaOH 0.1 M, 10 min) conditions, respectively, a 15% decrease and a 23% decrease in NAC content were observed. NAC content under moderate oxidative conditions (H_2_O_2_ 0.3%, 3 h) showed a 6% decrease. Several degradation products were observed during these experiments, and results are summarized in Table 3. Among degradation products, one peak (RT = 7.2–7.7 min) was predominant, and its area increased under all conditions (except basic conditions). As this product was suspected to be the NAC dimer, a standard solution of this compound was injected (known as Impurity C, Ph. Eur.). The similar retention time found validated our hypothesis.

### 2.3. Stability Study

Between their compounding and their administration, NAC infusion bags are stored in a refrigerator. Their stability was therefore evaluated under two conditions: 5 ± 3 °C (normal storage conditions) and 25 ± 2 °C (accelerated conditions).

As reported in the literature, stability was considered if NAC content remained higher than 90% of its initial content [11,12,13], but the results between 90% and 95% were also reported. Results are summarized in Table 4 and Table 5.

Either under 5 ± 3 °C or 25 ± 2 °C storage conditions, a statistically significant relationship between time and measured NAC content, % NAC dimer and pH was objectified. Despite slight variations due to analytical uncertainties, mean NAC content decreased over time. On the contrary, osmolality did not significantly vary over time (but also showed slight variations over time attributed to apparatus uncertainties). At T0, NAC content was found to be equal to 22.82 mg·mL^−1^. NAC content remained between 90% and 110% of the initial value (20.53–25.10 mg·mL^−1^) for 8 days under both storage conditions. However, NAC content fell below 95% of the initial value (21.68 mg·mL^−1^) after 4 days under refrigerated storage conditions (21.56 mg·mL^−1^), and after 3 days under ambient storage conditions (21.52 mg·mL^−1^). Moreover, percentage of NAC dimer became higher than 0.5% of NAC content after 7 days under refrigerated conditions, and after 2 days under ambient storage conditions. In the European Pharmacopoeia, NAC dimer (described as Impurity C) normal value for Active Pharmaceutical Ingredient (API) needs to be lower than 0.5% of NAC content [14].

Although this normal value is applied to API, we also decided to use it for our stability study of parenteral NAC, as normal values of API degradation products during stability studies are not reported in any pharmacopoeia and are only confidentially reported by pharmaceutical industries to sanitary agencies for marketing authorizations.

NAC dimerization was found to be the limiting parameter for NAC stability and its main degradation pathway. Therefore, we tried to decrease this phenomenon with different adjuvants. Mechanistically, thiols dimerization is favored by oxidative conditions. During our forced degradation studies, NAC dimer was multiplied by approximatively six when NAC was mixed with H_2_O_2_ 0.3% for 3 h.

We hypothesized that the adjunction of antioxidizing agents will be able to decrease NAC dimerization. As NAC need to be parenterally administered, antioxidizing agents were selected among commercial human IV medications. Percentage of NAC dimer was evaluated at T0, 4 days and 8 days, in triplicate, under refrigerated storage conditions in NAC syringes prepared in the same way than previously (with adjuvants adjunction as the sole difference). Adjuvants concentrations were chosen to be compatible with human administration, and to be easily made with commercial drugs. Results are summarized in Table 6.

Compared to control, sodium edetate, zinc gluconate and tocopherol reduced NAC dimerization, whereas ascorbic acid dramatically increased dimer content after reconstitution.

An addition of 125 µg·mL^−1^ zinc gluconate resulted in a 20% decrease in dimer content after 8 days, but 62.5 µg·mL^−1^ zinc gluconate addition was the second better adjuvant tested with a moderate 30% increase in dimer content after 8 days. As the total zinc intake by the patient was reduced twice with a theoretical effect able to increase NAC stability, 62.5 µg·mL^−1^ zinc gluconate was chosen for a new NAC stability study. This stability study was conducted under the same experimental procedure than previously. Moreover, a sterility test was validated according to European Pharmacopoeia recommendations, and sterility was evaluated at T0, day 4 and day 8 in a single experiment. Endotoxins were also evaluated at T0, day 4 and day 8 in a single experiment. Results are summarized in Table 7 and Table 8.

Although statistically significant relationships were highlighted (between time and especially NAC content and %NAC dimer) despite the addition of 62.5 µg/mL^−1^ zinc gluconate, NAC content remained higher than 95% of its initial value for 8 days under both storage conditions. Moreover, percentage of NAC dimer remained lower than 0.5% of NAC content for 4 days under ambient conditions, and more than 8 days under refrigerated storage conditions. Under refrigerated storage conditions, an unexplained moderate increase in NAC content was observed (lower than 10% of initial value). Such an increase in API concentration is often correlated to solvent evaporation [15]. Here, this increase cannot be attributed to diffusion of water vapor to the outside of the container, as it was not observed under the other experimental conditions.

## 3. Discussion

NAC quantification was performed by HPLC-UV with a validated stability-indicating dosing method. We studied the stability of 25 mg·mL^−1^ NAC diluted with D5W in PP/PA/PE bag.

Without adjuvants, the NAC content remained above 95% of its initial value for 2 days under ambient storage conditions, and for 3 days at 5 ± 3 °C, but also remained above 90% of its initial value for at least 8 days under both conditions. However, the percentage of dimeric NAC remained below 0.5% for only 1 day at room temperature and 4 days under refrigerated conditions. In the absence of normal values of degradation products, stability was established with respect to the NAC dimer content.

The stability of parenteral NAC has only been studied twice in the literature. Indeed, a concentration of 26 mg·mL^−1^ of NAC in D5W was found to be stable for 60 h at room temperature [5]. In this work, the NAC content decreased to less than 90% of its initial value at 72 h. Commercial vials of 20% NAC containing 0.5 mg·mL^−1^ edetate disodium had been used, leading to a final edetate content of 65 µg·mL^−1^. In our study, the parenteral formulation of NAC made from commercial parenteral NAC had a final edetate disodium content of 250 µg·mL^−1^, which could explain the differences. In another study, the NAC content of 60 mg·mL^−1^ in D5W, NS, or 0.45% sodium chloride was found to be greater than 90.0% of its initial value for 72 h at room temperature [6]. In this study, NAC content was not assessed after 72 h [6].

To improve the stability of parenteral NAC, several antioxidant agents were studied. Surprisingly, we found that ascorbic acid at 12.5 mg·mL^−1^ increased NAC dimerization. Although ascorbic acid is primarily known for its antioxidant properties, it can also act as a pro-oxidant agent. This paradoxical phenomenon has been previously described with thiol groups [16].

On the other hand, zinc gluconate at 125 µg·mL^−1^ was able to inhibit the dimerization of NAC, resulting in the most stable parenteral formulation. Moreover, the addition of 125 µg·mL^−1^ zinc gluconate resulted in a slight decrease in dimer content. Sufficient reducing agent may be able to reverse the dimerization, as the redox conversion reaction of thiol-disulfide is known to be reversible [17]. These results were confirmed by another stability study. 25 mg·mL^−1^ NAC + 125 µg·mL^−1^ zinc gluconate in D5W resulted in a 4-day stability of NAC when stored at room temperature and over 8 days under refrigerated conditions.

To our knowledge, our study was the first to examine the increased stability of parenteral ANC with adjuvants. Prior to opening, ready-to-use commercial pharmaceutical drugs containing excipients as stabilizers can have very long expiration dates. A wide variety of stabilizers have been described, and can be used to decrease for example oxidation, light degradation, or pH-catalyzed hydrolysis [18]. It may be useful in clinical practice to study the influence of these stabilizers to improve the stability of reconstituted drugs. Such studies require the expertise of physicians and pharmacists to choose the right excipient at the right dose and to avoid any negative impact on the patient.

Our study suffers from several limitations. First, due to a lack of specific equipment, we only evaluated visible particles, whereas for parenteral drugs, invisible particles should be evaluated. In addition, the ICH recommends studying the stability of products stored in a refrigerator under accelerated conditions at 25 °C/60% RH. In our study, only temperature was controlled, but humidity could also influence the results. We also evaluated only the chemical stability of NAC, but the stability of zinc gluconate could also be evaluated. In addition, NAC is a well-known chelating agent [19], and its interaction with zinc gluconate could yield to insoluble impurities and/or unexplained NAC content variations, such as those observed under 5 ± 3 °C storage conditions.

## 4. Materials and Methods

NAC analytical standard (Acetylcysteine, European Pharmacopoeia (Ph. Eur.) Reference Standard, EQDM) was used for method validation whereas NAC of pharmaceutical grade (Hidonac^®^ 5 g·25 mL^−1^, Zambon France SA, Issy-les-Moulineaux, France) was used for forced degradation study and stability studies. NAC was diluted in D5W (5% Dextrose Injection Ph. Eur., Viaflo^®^ 100 mL, Baxter SAS, Guyancourt, France). Parenteral ascorbic acid (Laroscorbine^®^ injection 1 g·5 mL^−1^, Bayer Healthcare SAS, Leverkusen, Germany), zinc gluconate (Zinc injectable 10 mg·mL^−1^, Aguettant, Lyon France), sodium edetate (Disodium edetate PE, Fagron, Thiais, France) and vitamin E (Tocopherol injection 100 mg·2 mL^−1^, Provepharm, Marseille, France) were used as adjuvants. The HPLC mobile phases were prepared using ultrapure water (HiPerSolv Chromanorm^®^, VWR International, Radnor, United States) and acetonitrile (HiPerSolv Chromanorm^®^, VWR International) of HPLC grade.

The mobile phase consisted of a mixture of acetonitrile and water (97:3, *v*/*v*) adjusted at pH = 3 with phosphoric acid. The mobile phase was filtered through a Millipore 0.45 µm cellulose filter and used in isocratic mode with a flow of 2 mL/min for 15 min. Wavelength was set at 200 nm, injection volumes were 10 µL and column (C18 XTerra^®^, 4.6 × 250 mm, 5 µm) temperature was set at 15 °C.

Volumes were aliquoted with a precision pipette (Thermo Scientific Finnpipette^®^ F2 500 µL, Thermo Fisher Scientific, Illkirch-Graffenstaden, France) and pH were measured with a Thermo Scientific Orion 4 Star^®^ pH-meter (Thermo Fisher Scientific, Illkirch-Graffenstaden, France), calibrated with Radiometer Analytical standard etalons^®^ (pH 4.005, pH 7.000 and pH 10.012). Osmolality was determined with Löser^®^ manual osmometer type 6 (Löser Messtechnik, Berlin, Germany), calibrated with 300 and 900 mOsm.kg^−1^ standard solutions.

The chromatographic method was carried out on an automatic high-performance liquid chromatography Dionex Ultimate 3000^®^ (Thermo Fisher Scientific, Illkirch-Graffenstaden, France) with a UV diode array detector. The apparatus was connected to an HP 1702 computer with chromatographic data processing software (Chromeleon^®^ Chromatography Management System, Version 6.80 SRH Biold 3161, 1994–2011 Dionex Corporation).

NAC solutions were prepared as follows: 30 mL were withdrawn from a 100 mL bag of D5W (5% Dextrose Injection Ph. Eur., Viaflo^®^, Baxter) using a sterile syringe. The Viaflo^®^ bag is a flexible plastic container made from a multilayer sheet (PL-2442) composed of polypropylene (PP), polyamide (PA) and polyethylene (PE). Then, 10 mL of NAC (Hidonac^®^ 5 g·25 mL^−1^) was introduced into the 100 mL Viaflo^®^ bag, yielding to a theoretical 25 mg·mL^−1^ NAC solution.

Stability study was conducted for 8 days, under two storages conditions: ambient temperature at 25 ± 2 °C and refrigerated conditions at 5 ± 3 °C. For each storage condition, five NAC solutions were prepared. NAC content and degradations products were evaluated in triplicate at T0, 6 h, day 1, day 2, day 3, day 4, day 7 and day 8. Osmolality, pH and the absence of visible particles were evaluated in a single experiment at the same sampling points.

Sterility test was validated according to European Pharmacopoeia with six reference strains (Biomérieux Bioball^®^): *Staphylococcus aureus* NCTC10788, *Bacillus subtilis* NCTC10400, *Pseudomonas aeruginosa* NCTC12924, *Candida albicans* NCPF3179, *Aspergillus braziliensis* NCPF2275 and *Clostridium sporogenes* NCTC532. Analyses were done under a Microbiological Safety Cabinet (MSC) (Herasafe^®^ KS, Thermo Scientific) with a Sterisart^®^ pump (Sartorius, Aubagne, France). 40 mL of NAC solution were aliquoted and added in 300 mL of sterile diluent + tween 80 (Becton Dickinson, Le Pont-de-Claix, France). Sterile diluent was filtered with Sterisart^®^ systems (Sartorius, 16477-GBD). Then, one container was filled with 100 mL of fluid thioglycollate medium (Becton Dickinson) and the second was filled with 100 mL of fluid TSA medium (Biomérieux, Marcy-l’Étoile, France). Samples were put in two incubators (Heratherm^®^ and Heraeus^®^ Thermo Scientific) for 14 days at 25 °C and 35 °C.

The endotoxin activities in the samples were determined by kinetic-turbidimetric assay, which is a commonly used variant of the limulus amebocyte lysate (LAL) test for quantifying endotoxins from Gram-negative bacteria and *cyanobacteria* in water samples. Pyrotell^®^T Limulus amebocyte Lysate (LAL) reagent is intended for the in vitro detection and quantification of endotoxin in the end-product testing of human injectable drugs, and medical devices. LAL is an aqueous extract of blood cells (amebocytes) from the horseshoe crab limulus polyphemus. In the presence of endotoxin, LAL becomes turbid and, under appropriate conditions, forms a solid gel-clot. The turbidimetric LAL test is performed by adding a given volume of Pyrotell^®^T to a given volume of specimen and incubating the reaction mixture at 37 °C. As a consequence, the turbidity in samples will increase in a dose- and time-dependent manner. The kinetic-turbidimetric assay quantifies the endotoxin activity based on the gel formation time to reach a designated turbidity (or a designated visible light absorption) by spectrophotometry. The higher the endotoxin concentration in the specimen, the faster turbidity will develop. The detection limit is 0.005 EU·mL^−1^.

Data were analyzed with a Microsoft Excel spreadsheet (Microsoft 365 MSO, Version 2210 Build 16.0.15726.20188). The relationship between the quantitative parameters explored in degradation studies (NAC content, % NAC dimer, osmolality and pH) and time was estimated with regression analysis and calculation of the coefficient of determination R^2^. Correlation coefficient tests at 95% confidence level were conducted between these four variables and time. *p* values lower than 0.05 were considered as statistically significant.

## 5. Conclusions

Without any adjuvant, 25 mg·mL^−1^ NAC diluted with D5W in polymeric Viaflo^®^ bag was stable 24 h at ambient temperature and for 4 days at 5 ± 3 °C. After addition of 62.5 µg·mL^−1^ zinc gluconate, the resulting stability was increased to 4 days at ambient temperature and to 8 days at 5 ± 3 °C.

Short-term storage length of reconstituted drugs can be increased by an adequate stabilizer adjunction. Such a mixture must maintain patient innocuity and needs to be evaluated for both its chemical and microbiological stability and patient tolerance.

## Figures and Tables

**Figure 1 pharmaceuticals-16-00072-f001:**
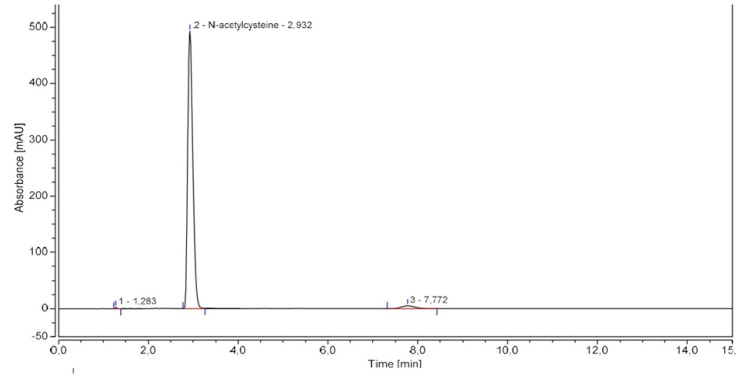
Representative NAC chromatogram.

**Table 1 pharmaceuticals-16-00072-t001:** Repeatability, intermediate precision, and accuracy.

Samples	Repeatability (% RSD within-Day)	Intermediate Precision (% RSD between-Day)	Accuracy (Bias in %)
NAC 450 µg·mL^−1^	0.640%	2.458%	−1.827%
NAC 500 µg·mL^−1^	0.287%	1.909%	−1.656%
NAC 550 µg·mL^−1^	0.319%	0.910%	−2.081%

**Table 2 pharmaceuticals-16-00072-t002:** Robustness.

Conditions		%RSD ^1^
pH of mobile phase	2.7	0.887
	3.0	0.347
	3.3	2.757
Composition of mobile phase	98:2	0.501
	97:3	0.347
	96:4	0.281
Flow	1.8 mL·min^−1^	0.216
	2.0 mL·min^−1^	0.347
	2.2 mL·min^−1^	0.244
Column	C18 Kromasil^®^, 4.6 × 250 mm, 5 µm	0.089
	C18 XTerra^®^, 4.6 × 250 mm, 5 µm	0.347
	C18 Lichrospher^®^, 4.6 × 250 mm, 5 µm	4.404
Wavelength	198 nm	1.502
	200 nm	0.347
	202 nm	0.588

^1^ Relative Standard Deviation.

**Table 3 pharmaceuticals-16-00072-t003:** N acetylcysteine forced degradation study.

Experimental Conditions	API Degradation	Degradation Products’ Retention Times
Heat (80 °C, 3 h)	24%	1.3 min, 1.5 min, 1.8 min, 2.2 min, 5.3 min, 7.2–7.7 min
Light (sunlamp, 28 days)	3%	7.2–7.7 min
Oxidation (H_2_O_2_ 0.3%, 3 h)	6%	1.6 min, 2.0 min, 7.2–7.7 min
Acid (HCl 0.5 M, 1 min)	15%	1.5 min, 1.7 min, 7.2–7.7 min
Alkaline (NaOH 0.1 M, 10 min)	23%	1.9 min, 2.0 min, 7.2–7.7 min

**Table 4 pharmaceuticals-16-00072-t004:** NAC stability study, 5 ± 3 °C storage conditions.

	T0	6 h	1 Day	2 Days	3 Days	4 Days	7 Days	8 Days	R ^2,4^	*p* Value ^5^
NAC content ± S.D. ^1^	22.82 ± 1.99	22.28 ± 0.18	22.09 ± 0.04	22.41 ± 0.41	22.22 ± 0.35	21.56 ± 0.23	21.88 ± 0.21	21.42 ± 0.50	0.65	<0.02
% NAC dimer ± S.D. ^2^	0.20 ± 0.03	0.29 ± 0.01	0.32 ± 0.01	0.34 ± 0.02	0.37 ± 0.01	0.42 ± 0.01	0.53 ± 0.02	0.54 ± 0.04	0.94	<0.001
Osmolality ^3^	511	500	510	508	524	510	515	512	0.15	NS ^6^
pH	6.26	6.24	6.24	6.21	6.24	6.15	6.10	6.12	0.87	<0.001
Visible particles	none	none	none	none	none	none	none	none		

^1^ expressed as the mean value of triplicate analysis, in mg·mL^−1^; ^2^ expressed as the mean value of triplicate analysis; ^3^ expressed in mOsm.kg^−1^; ^4^ coefficient of determination; ^5^ correlation coefficient test significance; ^6^ not statistically significant.

**Table 5 pharmaceuticals-16-00072-t005:** NAC stability study, 25 ± 2 °C ambient storage conditions.

	T0	6 h	1 Day	2 Days	3 Days	4 Days	7 Days	8 Days	R ^2,4^	*p* Value ^5^
NAC content ± S.D. ^1^	22.82 ± 1.99	22.15 ± 0.17	21.78 ± 0.26	22.03 ± 0.43	21.52 ± 0.29	21.95 ± 0.25	21.61 ± 0.42	21.27 ± 0.20	0.58	<0.05
% NAC dimer ± S.D. ^2^	0.20 ± 0.03	0.29 ± 0.01	0.39 ± 0.02	0.52 ± 0.01	0.61 ± 0.01	0.71 ± 0.02	1.09 ± 0.10	1.13 ± 0.18	0.99	<0.001
Osmolality ^3^	511	508	507	507	505	521	515	520	0.48	NS ^6^
pH	6.26	6.13	6.11	6.04	6.05	6.04	6.04	5.95	0.65	<0.02
Visible particles	none	none	none	none	none	none	none	none		

^1^ expressed as the mean value of triplicate analysis, in mg·mL^−1^; ^2^ expressed as the mean value of triplicate analysis; ^3^ expressed in mOsm.kg^−1^; ^4^ coefficient of determination; ^5^ correlation coefficient test significance; ^6^ not statistically significant.

**Table 6 pharmaceuticals-16-00072-t006:** Influence of antioxidizing agents on NAC dimerization among time.

		T0	4 Days	8 Days
Control	% NAC dimer	0.20	0.42	0.54
	variation from baseline		×2.1	×2.7
ascorbic acid 12.5 mg·mL^−1^	% NAC dimer	0.27	2.08	3.68
	variation from baseline		×7.7	×13.6
Sodium edetate 2 mg·mL^−1^	% NAC dimer	0.39	0.50	0.64
	variation from baseline		×1.3	×1.6
Zinc gluconate 12.5 µg·mL^−1^	% NAC dimer	0.36	0.48	0.58
	variation from baseline		×1.3	×1.6
Zinc gluconate 62.5 µg·mL^−1^	% NAC dimer	0.28	0.33	0.36
	variation from baseline		×1.2	×1.3
Zinc gluconate 125 µg·mL^−1^	% NAC dimer	0.31	0.29	0.25
	variation from baseline		×0.9	×0.8
Tocopherol 1.25 mg·mL^−1^	% NAC dimer	0.28	0.46	0.63
	variation from baseline		×1.6	×2.3

**Table 7 pharmaceuticals-16-00072-t007:** NAC + 62.5 µg·mL^−1^ zinc gluconate stability study, 5 ± 3 °C storage conditions.

	T0	1 Day	2 Days	3 Days	4 Days	7 Days	8 Days	R ^2,4^	*p* Value ^5^
NAC content ± S.D. ^1^	21.78 ± 0.20	21.97 ± 0.43	22.33 ± 0.60	22.42 ± 0.61	22.17 ± 0.10	22.57 ± 0.13	23.25 ± 0.05	0.80	<0.01
% NAC dimer ± S.D. ^2^	0.28 ± 0.03	0.28 ± 0.03	0.30 ± 0.04	0.32 ± 0.03	0.34 ± 0.01	0.36 ± 0.01	0.36 ± 0.01	0.92	<0.001
Osmolality ^3^	499	512	505	502	526	522	484	0.01	NS ^6^
pH	5.22	5.14	5.19	5.19	5.21	5.14	5.20	0.03	NS ^6^
Sterility	Sterile	ND	ND	ND	Sterile	ND	Sterile		
Endotoxins ^4^	0.1<	ND	ND	ND	0.1<	ND	0.1<		
Visible particles	none	none	none	none	none	none	none		

^1^ expressed as the mean value of triplicate analysis, in mg·mL^−1^; ^2^ expressed as the mean value of triplicate analysis; ^3^ expressed in mOsm.kg^−1^; ^4^ expressed in IU.mL^−1^; ^4^ coefficient of determination; ^5^ correlation coefficient test significance; ^6^ not statistically significant.

**Table 8 pharmaceuticals-16-00072-t008:** NAC + 62.5 µg·mL^−1^ zinc gluconate stability study, 25 ± 2 °C ambient storage conditions.

	T0	1 Day	2 Days	3 Days	4 Days	7 Days	8 Days	R ^2,4^	*p* Value ^5^
NAC content ± S.D. ^1^	22.10 ± 0.20	21.93 ± 0.18	21.40 ± 0.15	21.58 ± 0.08	21.65 ± 0.10	21.17 ± 0.08	21.23 ± 0.18	0.75	<0.02
% NAC dimer ± S.D. ^2^	0.27 ± 0.03	0.29 ± 0.03	0.34 ± 0.03	0.38 ± 0.04	0.42 ± 0.05	0.50 ± 0.04	0.53 ± 0.04	0.99	<0.001
Osmolality ^3^	541	540	580	584	569	540	538	0.06	NS ^6^
pH	5.29	5.29	5.31	5.32	5.33	5.41	5.43	0.96	<0.001
Sterility	Sterile	ND	ND	ND	Sterile	ND	Sterile		
Endotoxins ^4^	0.1<	ND	ND	ND	0.1<	ND	0.1<		
Visible particles	none	none	none	none	none	none	none		

^1^ expressed as the mean value of triplicate analysis, in mg·mL^−1^; ^2^ expressed as the mean value of triplicate analysis; ^3^ expressed in mOsm.kg^−1^; ^4^ expressed in IU.mL^−1^; ^4^ coefficient of determination; ^5^ correlation coefficient test significance; ^6^ not statistically significant.

## Data Availability

Not applicable.

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
