# Peer review of "Stability Study of Parenteral N-Acetylcysteine, and Chemical Inhibition of Its Dimerization"

_pharmaceuticals, 2023, doi:10.3390/ph16010072_

Round 1
Reviewer 1 Report
Some questions about method validation:
1. I don't see the relevant data of the linear survey;
2. Why is the range of repeatability, intermediate precision and precision only 90% - 110%. The labeled amount of the preparation usually from 90.0% to 110.0%. Considering the risk and the applicability of the method, it is more appropriate to conduct the inspection in the concentration range of 80.0% - 120.0%.
3. The conditions selected for forced degradation are not appropriate. The sample degradation amount should be between 5% and 20%. However, excessive destruction of the sample will produce secondary degradation products that will not be produced during the drug stability study and under normal destruction conditions, which is not the purpose of the degradation test.
4. Whether the author has conducted relevant durability validation, such as different column temperature and flow rate?However, adequate method validation is the basis for accurate determination.
Author Response
- I don't see the relevant data of the linear survey
We thank the reviewer for this remark, as we forgot this point, and moreover a small typo was done (we tested seven concentrations, not five).
P2L53 : « Linearity of HPLC-UV standard curve for NAC was determined with five concentrations ranging from 125 to 1500 µg.mL-1 prepared in sextuplicate » was replaced by « Linearity of HPLC-UV standard curve for NAC was determined with seven concentrations ranging from 125 to 1500 µg.mL-1 prepared in sextuplicate. Linearity was proved between 250 and 1500 µg.mL-1 (R2 > 0.99 and deviation < 10% for each concentration). »
- Why is the range of repeatability, intermediate precision and precision only 90% - 110%. The labeled amount of the preparation usually from 90.0% to 110.0%. Considering the risk and the applicability of the method, it is more appropriate to conduct the inspection in the concentration range of 80.0% - 120.0%.
We agree with reviewer. As specified by ICH, « The specified range is normally derived from linearity studies ». Linearity was studied between 250 and 1500 µg.mL-1, which cover the range of 80.0 – 120.0%. We focused on analytical uncertainty between 90 and 110% to cover the classical range of stability studies (the main objective of our work).
- The conditions selected for forced degradation are not appropriate. The sample degradation amount should be between 5% and 20%. However, excessive destruction of the sample will produce secondary degradation products that will not be produced during the drug stability study and under normal destruction conditions, which is not the purpose of the degradation test.
Forced degradation studies were reproduced under new conditions to attain the requested degradation.
The manuscript, table 3 and supplementary material were modified with new results.
- Whether the author has conducted relevant durability validation, such as different column temperature and flow rate?However, adequate method validation is the basis for accurate determination.
We conducted a robustness study, with variations on several parameter, as suggested by reviewer.
P2L57 : « A robustness study was also conducted with slight variations on several parameters (mobile phase composition and pH, flow, wavelength and column). pH increase of mobile phase and column furnisher were shown to have an influence on reliability of the results (%RSD > 2.0%).» was added.
Table 2. was added
Reviewer 2 Report
The manuscript submitted for review concerns stability studies of parenteral N-acetylcysteine. Stability testing of a drug is one of the mandatory quality control tests, the primary purpose of which is to determine the expiry date and the efficacy and safety of the drug throughout its shelf life. Although the topic addressed by the authors is not new, I believe it is important in terms of the development of this form of the drug. However, I suggest that the quality of the article should be significantly improved:
1. Introduction.
a. I suggest including information on drug stability studies. This will allow the potential reader to familiarise themselves with current guidelines for this type of testing.
b. Please describe and justify the choice of antioxidants used in the studies. Why they were used in these doses and not in others.
2. Materials and Methods.
c. Stability studies should be performed based on ICH guidelines (Q1A-Q1F), WHO Annex 2. This information is also included in the papers cited by the authors. According to the ICH guidelines, testing under controlled conditions in climate chambers is required:
products stored at room temperature: under long-term conditions (25˚C/60%RH), under intermediate conditions (30˚C/65%RH), under accelerated conditions (40˚C/75%RH), at elevated temperatures using the Arrhenius equation;
products stored in the refrigerator: under long-term conditions (5˚C), under accelerated conditions (25˚C/60%RH).
The authors did not include any information on which of the recommended stability analysis methods they used.
d. According to pharmacopoeial recommendations, if no insoluble impurities are found, the liquid should be tested by microscopy or instrumental methods.
I suggest testing according to ICH and Pharmacopoeia guidelines.
e. There is no information on the equipment used in the study (e.g. osmometer).
f. There is no statistical analysis provided.
3. Results.
g. Table 2 is illegible. Degradation products' retention Times ? Please place an explanation below the table.
h. Lines 86-98 should be in the 'Materials and Methods' section.
i. To increase the readability of the quoted temperatures I suggest that the "+/ -" designations be changed to ±. On the other hand, "+2/+8 oC" ?
j. Table 3. NAC stability study, +2/+8 oC storage conditions. At what temperature was the study performed?
k. Please provide standard deviation values in the tables. Standard deviation is the primary measure of variability in results subjected to direct observation.
l. Table 4. NAC stability study, ambient storage conditions. For better readability, please indicate the temperature value of the study.
m. Table 3 and 4. Please explain why the NAC content alternately decreased and increased over time, instead of gradually decreasing over time.
n. Table 6. NAC content increased over time. Please explain.
o. Why did the osmotic pressure values change significantly over time? Please explain.
p. Annotations under the tables should be separated by a comma or semicolon.
Author Response
- Introduction
- I suggest including information on drug stability studies. This will allow the potential reader to familiarise themselves with current guidelines for this type of testing.
P2L45 : a paragraph was added in the introduction, according to reviewer recommendation:
“Beyond-Use-Date of commercial drugs must be established with a stability study de-signed as required by the scientific guideline established by the International Council for Harmonization (ICH) [8], using a dosing method validated to be stability-indicating [9]. A stability-indicating method was defined as a validated quantitative analytical method that can detect the changes with time in the properties of the drug product without inter-ference [10].”
References [8], [9] and [10] (ICH and FDA guidelines) were added.
- Please describe and justify the choice of antioxidants used in the studies. Why they were used in these doses and not in others.
P2L49: a paragraph was added in the introduction, according to reviewer recommendation:
“Our preliminary results showed that NAC major degradation product yielded from NAC oxidation. There is a wide variety of available antioxidizing agents, but to avoid high risk compounding steps with raw material, we decided to select only antioxidizing agents available as ready-to-use commercial drugs: ascorbic acid, zinc gluconate and tocopherol. Doses were tried first to correspond to the adjunction of one commercial drug vial to NAC to test the higher antioxidant property as possible. Disodium edetate was also tested at 2 mg.mL-1 as undiluted commercial NAC drug contain this excipient at the same concen-tration. From these results, we planned to find the best antioxidizing agent and next to de-crease dose to determine the lowest dose as possible to be compatible with human ad-ministration.”
- Materials and Methods.
- Stability studies should be performed based on ICH guidelines (Q1A-Q1F), WHO Annex 2. This information is also included in the papers cited by the authors. According to the ICH guidelines, testing under controlled conditions in climate chambers is required:
products stored at room temperature: under long-term conditions (25˚C/60%RH), under intermediate conditions (30˚C/65%RH), under accelerated conditions (40˚C/75%RH), at elevated temperatures using the Arrhenius equation;
products stored in the refrigerator: under long-term conditions (5˚C), under accelerated conditions (25˚C/60%RH).
The authors did not include any information on which of the recommended stability analysis methods they used.
P3L85: “Between their compounding and their administration, NAC infusion bags are stored in a refrigerator. Their stability was therefore evaluated under two conditions: 5 ± 3 oC (normal storage conditions) and 25 ± 2 oC (accelerated conditions).” was added.
As our study was not realized in climatic chambers, we also added a short paragraph in the limitations part: P7L213: “Moreover, the ICH recommend to study stability of products stored in the refrigerator un-der accelerated conditions at 25 oC / 60 % RH. During our study, only temperature was controlled, but the hygrometry could also influence the results.”
- According to pharmacopoeial recommendations, if no insoluble impurities are found, the liquid should be tested by microscopy or instrumental methods.
I suggest testing according to ICH and Pharmacopoeia guidelines.
We totally agree with the reviewer, but both the methods detailed in the Pharmacopoeia involve material (particle counter or specific filtration device) we don’t have in our laboratory.
P7L212, we have mentioned this bia in the limitations part: First, due to a lack of specific material, we only evaluated visible particles whereas for parenteral drugs, invisible particles must be evaluated.
- There is no information on the equipment used in the study (e.g. osmometer).
“Osmolality was determined with Löser® manual osmometer type 6, calibrated with 300 and 900 mOsm.kg-1 standard solutions.” was added P7L235
- There is no statistical analysis provided.
A statistical analysis was realized and the manuscript was modified as follow:
Tables 3, 4, 6 and 7 were modified to include statistical analysis (in two last columns)
P4L111: “Either under 5 ± 3 oC or 25 ± 2 oC storage conditions, a statistically significant relationship between time and measured NAC content, % NAC dimer and pH was objectified. » was added.
P6L166: “When 62.5 µg.mL-1 zinc gluconate was added, NAC content remained higher than 95 % of its initial value for 8 days under both storage conditions. Moreover » was replaced by « Although statistically significant relationships were highlighted (between time and especially NAC content and %NAC dimer) despite the addition of 62.5 µg/mL-1 zinc gluconate, NAC content remained higher than 95 % of its initial value for 8 days under both storage conditions. Moreover, »
P8L265:” Data were analyzed with a Microsoft Excel spreadsheet (Microsoft 365 MSO, Version 2210 Build 16.0.15726.20188). The relationship between the quantitative parameters explored in degradation studies (NAC content, % NAC dimer, osmolality and pH) and time was estimated with regression analysis and calculation of the coefficient of determination R2. Correlation coefficient tests at 95% confidence level were conducted between these 4 variables and time. P values lower than 0.05 were considered as statistically significant.” was added.
A section “Acknowledgments” was added.
- Table 2 is illegible. Degradation products' retention Times ? Please place an explanation below the table.
P3L73: “The last column of table 2 presents the retention times of degradation products (each retention time correspond to one degradation product). Observed chromatograms can be found in supplementary material.” was added.
- Lines 86-98 should be in the 'Materials and Methods' section.
This paragraph was moved in Materials and methods
- To increase the readability of the quoted temperatures I suggest that the "+/ -" designations be changed to ±. On the other hand, "+2/+8 oC" ?
In the manuscript, « +2/+8 oC was changed to 5 ± 3 oC »
- Table 3. NAC stability study, +2/+8 oC storage conditions. At what temperature was the study performed?
This study was performed at 5 ± 3 oC. This temperature correspond to the ICH recommendation (Drug substances intended for storage in a refrigerator, storage condtions 5 ± 3 oC)
- Please provide standard deviation values in the tables. Standard deviation is the primary measure of variability in results subjected to direct observation.
Standard deviations were added for NAC content and NAC dimer content in Tables.
NAC content CI95 were suppressed as they are very close to SD, and the two values do not provide additional informations.
- Table 4. NAC stability study, ambient storage conditions. For better readability, please indicate the temperature value of the study.
“25 ± 2 oC” was added in the title of Tables 4 and 7
- Table 3 and 4. Please explain why the NAC content alternately decreased and increased over time, instead of gradually decreasing over time.
P4L111: “Despite slight variations due to analytical uncertainties mean NAC content decreased over time. On the contrary, osmolality did not significantly vary over time (but also showed slight variations over time attributed to apparatus uncertainties). » was added
- Table 6. NAC content increased over time. Please explain.
We apologize, but we have no explanation for this phenomenon.
A sentence was added P6L171 after “Under refrigerated storage conditions, an unexplained moderate increase in NAC content was observed (lower than 10% of initial value).”
“ This increase cannot be attributed to diffusion of water vapour to the outside of the container, as it was not observed under the other experimental conditions.”
- Why did the osmotic pressure values change significantly over time? Please explain.
Statistical analysis showed us that osmolality did not significantly vary in our study.
P4L111 : « On the contrary, osmolality did not significantly vary over time (but also showed slight variations over time). » was added.
- Annotations under the tables should be separated by a comma or semicolon.
Semicolons were added between annotations
Reviewer 3 Report
Comments: The paper is about stability of N-acetylcysteine. Most of places grammatical mistakes are there. Some mistakes are discussed below:
Line 32: influenza
Table 1: % RSD values are very low. Similarly, the accuracy values are low. You can check, have you calculated accurately. For formulas of precision and accuracy, you can go through the paper: 10.1016/j.foodchem.2013.11.085
The values in Table 2 and its written portion were different?
Similarly the values of Table 3 and 4 are not matching to the values represented in written portion.
Zinc gluconate 125 μ g.mL-1 Showed less NAC dimer formation than Zinc gluconate 62.5 μ g.mL-1 than why you use lower concentration.
The result part and the discussion was poorly written.
Tha paper should be word by word accurately checked by authors to reduce the mistakes.
Author Response
Reviewer 3.
Line 32: influenza
P1L32, influenza was corrected
Table 1: % RSD values are very low. Similarly, the accuracy values are low. You can check, have you calculated accurately. For formulas of precision and accuracy, you can go through the paper: 10.1016/j.foodchem.2013.11.085
%RSD and accuracy values were checked, results displayed are OK.
The values in Table 2 and its written portion were different?
A typo was corrected in the written portion (for H2O2 conditions, we reported in the Table the last experiment, and in the written portion another one).
Similarly the values of Table 3 and 4 are not matching to the values represented in written portion.
P4L114 « after 3 days under refrigerated storage conditions, and after 2 days under 114 ambient storage conditions. Moreover, percentage of NAC dimer became higher than 115 0.5% of NAC content after 4 days under refrigerated conditions, and after one day under 116 ambient storage conditions. » was replaced by : « after 4 days under refrigerated storage conditions (21.56 mg.mL-1), and after 3 days under ambient storage conditions (21.52 mg.mL-1). Moreover, percentage of NAC dimer became higher than 0.5% of NAC content after 7 days under refrigerated conditions, and after 2 days under ambient storage conditions »
Moreover, P6L180 « became higher than 0.5 % after 1 day at ambient temperature, and after 4 180 days under refrigerated conditions » was replaced by: “However, percentage of NAC dimer remained lower than 0.5 % only 1 day at ambient temperature, and 4 days under refrigerated conditions”
Zinc gluconate 125 μg.mL-1 Showed less NAC dimer formation than Zinc gluconate 62.5 μ g.mL-1 than why you use lower concentration.
We agree with reviewer, but we decided to test the lowest adjuvant concentration as possible (for being compatible with patients’ administration). As the sentence was not clear enough, we slightly modified it:
P5L147 “As the total zinc intake by the patient was reduced twice with a theoretical effect able to increase NAC stability, this adjuvant was chosen for a new NAC stability study” was replaced by “As the total zinc intake by the patient was reduced twice with a theoretical effect able to increase NAC stability, 62.5 µg.mL-1 zinc gluconate was chosen for a new NAC stability study.”
The result part and the discussion was poorly written. Tha paper should be word by word accurately checked by authors to reduce the mistakes.
The discussion L173-L215 was checked and corrected
Several modifications and corrections were also done:
L25 : « also » was deleted
L32 : « in dextrose 5% in water (D5W) » was added
L39 : « normal saline (NS) » was added
L47 : « first » was deleted
L48 « to try » was replaced by « we tried »
L131 « parenterally » was deleted
L227 « in » was replaced by « of »
L237 « linked » was deleted
Table 5 : « X » were replaced by « × »
Tables 6 and 7: CI95 were corrected as they were incorrectly calculated in previous version.
Round 2
Reviewer 1 Report
Accept
Author Response
Reviewer 1
We thanks reviewer 1, and our manuscript was checked again.
Reviewer 2 Report
The authors have significantly improved their manuscript. Unfortunately, two important issues are still not clarified. The presence of insoluble impurities was not investigated. Perhaps this analysis would have helped to explain the anomalies of NAC content during stability studies.
I also suggest explaining the increase in NAC over time (currently Table 7) based on the literature analysis. Has another article(s) described a similar relationship?
Author Response
Reviewer 2
The presence of insoluble impurities was not investigated. Perhaps this analysis would have helped to explain the anomalies of NAC content during stability studies. I also suggest explaining the increase in NAC over time (currently Table 7) based on the literature analysis. Has another article(s) described a similar relationship?
We agree with both reviewer’s remarks.
As reported P8L244, visible particles contamination was investigated during our study. A short sentence about the need to evaluated invisible particles was added, but our laboratory does not have liquid particles counter.
P7L199: “ Under refrigerated storage conditions, an unexplained moderate increase in NAC content was observed (lower than 10% of initial value). This increase cannot be attributed to diffusion of water vapour to the outside of the container, as it was not observed under the other experimental conditions.” was replaced by “Under refrigerated storage conditions, an unexplained moderate increase in NAC content was observed (lower than 10% of initial value). Such an increase in API concentration is often correlated to solvent evaporation [15]. Here, this increase cannot be attributed to diffusion of water vapour to the outside of the container, as it was not observed under the other experimental conditions.”
Ref 15 was added (González-González, O.; Ramirez, I.O.; Ramirez, B.I.; O'Connell, P.; Ballesteros, M.P.; Torrado, J.J.; Serrano, D.R. Drug Stability: ICH versus Accelerated Predictive Stability Studies. Pharmaceutics 2022, 14, 2324.)
P8L249: “In addition, NAC is a well-known chelating agent [19], and its interaction with zinc glu-conate could yield to insoluble impurities and / or unexplained NAC content variations such as those observed under 5 ± 3 oC storage conditions.” was added.
Ref 19 was added (Flora, S.J. Structural, chemical and biological aspects of antioxidants for strategies against metal and metalloid exposure. Oxid Med Cell Longev 2009, 2, 191-206.)